# An Initial Study on the Use of Machine Learning and Radio Frequency Identification Data for Predicting Health Outcomes in Free-Range Laying Hens

**DOI:** 10.3390/ani13071202

**Published:** 2023-03-30

**Authors:** Mitchell Welch, Terence Zimazile Sibanda, Jessica De Souza Vilela, Manisha Kolakshyapati, Derek Schneider, Isabelle Ruhnke

**Affiliations:** 1School of Science & Technology, University of New England, Armidale, NSW 2351, Australia; 2Precision Agriculture Research Group, University of New England, Armidale, NSW 2351, Australia; 3School of Environmental and Rural Science, University of New England, Armidale, NSW 2351, Australia

**Keywords:** poultry, aviary, eggs, welfare, smart farming, big data

## Abstract

**Simple Summary:**

Maintaining the health and welfare of laying hens is essential to ensure optimal productivity and to build consumer confidence. Free-range egg production systems present unique challenges, including increased exposure to parasites, infection, and injury. The ability to predict and prevent these health problems translates into significant financial savings. This study examines the use of Radio Frequency Identification (RFID) to measure the movement behavior of laying hens and machine learning to forecast individual hens’ health status. The machine learning workflow incorporates data resampling and important feature identification to overcome the highly unbalanced dataset. Results indicate an average of 28% Spotty Liver Disease, 33% round worm, and 33% tape worm infection correctly predicted by the end of the production period. The monitoring of hens’ health during the early laying period can lead to similar performances in predicting infections compared to models trained with peak laying data. Future research can improve the initial predictions by incorporating additional data streams to provide a more comprehensive view of flock health.

**Abstract:**

Maintaining the health and welfare of laying hens is key to achieving peak productivity and has become significant for assuring consumer confidence in the industry. Free-range egg production systems represent diverse environments, with a range of challenges that undermine flock performance not experienced in more conventional production systems. These challenges can include increased exposure to parasites and bacterial or viral infection, along with injuries and plumage damage resulting from increased freedom of movement and interaction with flock-mates. The ability to forecast the incidence of these health challenges across the production lifecycle for individual laying hens could result in an opportunity to make significant economic savings. By delivering the opportunity to reduce mortality rates and increase egg laying rates, the implementation of flock monitoring systems can be a viable solution. This study investigates the use of Radio Frequency Identification technologies (RFID) and machine learning to identify production system usage patterns and to forecast the health status for individual hens. Analysis of the underpinning data is presented that focuses on identifying correlations and structure that are significant for explaining the performance of predictive models that are trained on these challenging, highly unbalanced, datasets. A machine learning workflow was developed that incorporates data resampling to overcome the dataset imbalance and the identification/refinement of important data features. The results demonstrate promising performance, with an average 28% of Spotty Liver Disease, 33% round worm, and 33% of tape worm infections correctly predicted at the end of production. The analysis showed that monitoring hens during the early stages of egg production shows similar performance to models trained with data obtained at later periods of egg production. Future work could improve on these initial predictions by incorporating additional data streams to create a more complete view of flock health.

## 1. Introduction

Animal health and welfare plays a key role in both optimization of livestock production and the perceptions of consumers that are driving the increasing demand for ethically-sourced agriculture products [1]. This consumer drive has had a significant impact on the poultry industry, leading to an increased demand for free-range eggs, and a subsequent response by the industry towards production systems that allow hens to experience greater mobility and freedom to move. Free-range production systems allow hens access to open ranges, floor spaces, dust bathing material, aviary tiers, nest boxes, and water lines [2,3]. While the adoption of such production systems addresses the concerns of consumers regarding animal welfare, they also expose the hens to a range of additional health challenges that must be carefully managed to avoid undermining productivity [4,5,6]. This diverse environment (coupled with individual preferences) can expose individual hens and sub-populations to various levels of biosecurity risks, inter-hen stressors, and collision damage [7]. Typical welfare concerns that can be seen within free-range flocks, which are associated with production losses, include diseases such as Spotty Liver Disease (SLD) [8], Fatty Liver Hemorrhagic Syndrome (FLHS) [9], and intestinal parasites such as *Ascaridia galli* (*A. galli*) [10] and various cestodes [11], along with injuries related to movement and social activities such as plumage or beak damage [12]. While changes in behavior and levels of activity can be observed accompanying the aforementioned conditions and parasites, there have been relatively few attempts to develop monitoring approaches capable of predicting the onset of health issues based around individual-level behavioural monitoring.

Monitoring the health of laying hens has been traditionally achieved at the flock-level, using the data derived from the production system such as average number of eggs per hen, average weights, average feed intake, average water intake, and average egg quality measures. These flock-level metrics have been used to model egg production curves [13], with studies such as [14] applying machine learning algorithms to detect early changes that indicate future production losses. Colles et al. [15] have extended the idea of early detection by demonstrating the use of computer vision and optical flow across the flock movement for the early warning of SLD and presence of the causative bacteria. Similarly, the combination of audio sensing and machine learning has been applied to develop a workflow capable of successfully identifying abnormal sounds linked with disease in broiler production [16]. These approaches provide a high-level view for flock management but do not capture details required for predictions at the individual hen level or vulnerable flock sub-populations [17]. Machine learning has seen wide use in animal production studies due to the availability of inexpensive sensing hardware and large data sets. It has been successfully applied to improve welfare and production outcomes, and continues to be a driver for innovation across the sector [18,19].

Radio Frequency Identification (RFID) technology has been widely adopted as a viable approach for identifying and tracking the movement of individual animals in agricultural production systems, both within scientific studies and commercial systems [20,21,22,23,24,25,26,27]. The low cost and the small size of on-animal devices makes this technology a versatile solution for investigating how animals access resources and use areas within a production system. RFID technology of this type has been used in a number of studies to track the weight, feeding behavior, and oviposition of hens using specialized equipment within aviary systems and feed chains [28,29]. This work has been applied to free-range production systems [30,31,32] for monitoring range usage, along with movements within feeding the system by deploying a comprehensive system of RFID receivers and antennas throughout the open range and indoor housing. This concept was extended in [30], where unsupervised machine learning approaches were applied to investigate the existence of sub-populations based upon high-level production system usage patterns of free-range hens.

This study builds upon this initial body of work by applying a supervised machine learning algorithm to predict the presence of diseases SLD and FLHS, the parasites *A. galli* and cestodes, and injuries such as beak, plumage, or keel bone damage based upon range and aviary usage patterns across the production lifecycle of RFID-tagged free-range hens. This combination of RFID technology and machine learning for individual-level forecasting of health conditions in laying hens represents a novel application of the technologies aimed at improving production through early intervention. The remaining sections present an analysis of the underlining dataset, aimed at understanding its suitability for training machine learning classifiers, a machine learning workflow that incorporates a dataset balancing algorithm, and evaluation of a classifier algorithm trained on the RFID data for predicting the presence of altered hen movement, which serves as an indicator for these aforementioned health conditions.

## 2. Materials and Methods

### 2.1. Data Collection and Processing

Data were collected from five commercial free-range flocks of Lohmann Brown laying hens monitored from December 2016 until February 2019. Each flock contained 40,000 hens, from which 3125 hens were randomly selected and fitted with leg bands containing Monza R6 UHF RFID transponders (supplied by Impinj, Seattle, WA, USA). This resulted in 15,625 hens that were initially monitored across the five flocks. The housing system was furnished with a 3-tier aviary system that was instrumented with custom-made antennae installed laterally along the entry of the nest boxes, the feeder chain systems (upper and lower feeders), and on the partitioned range section. The full details on the layout and antenna configuration are presented in Sibanda et al., 2019 [33]. In brief, to organize and categorize the movement activities of hens, the number of hen visits was determined by counting the number of antennae registrations detected for at least 10 s, and the duration spent on each antenna was summed up to calculate the time spent in nest boxes, feeders and outdoor range each day. Utilizing the RFID data on hen movements, we computed the mean daily durations that individual hens spent on different resources such as the upper feeder, lower feeder, nest boxes, and range. These calculations were conducted for four specific laying periods, namely the pre-laying (PRL) period (18–22 weeks), peak laying (PL) period (23–33 weeks), late laying (LL) period (34–54 weeks), and end of laying (EL) period (55–74 weeks), to capture variations in hen behavior throughout their production life. This produced a feature data set that contained 16 features for each hen surviving until the end of each flock’s production life. Subsequently, we utilized these features as predictors in our machine learning models to accurately classify and predict the health conditions of the hens based on their individual data.

At 74 weeks of age, all flocks were depopulated, and the remaining tagged hens were necropsied according to the procedure outlined in [34] to assess the health and welfare conditions. From this assessment, complete datasets for 9362 hens were available (i.e., due to lost/malfunctioning tags and flock mortality) and the conditions chosen for the responses (targets) used for predictive modelling are outlined in Table 1. Response variables were converted from their original (multi-class) domains into binary data outputs to simplify the modelling process and alleviate the level of class imbalance (for example, the three keel bone damage classes outlined in [34] were simplified to two classes, with scores of 1 and 2 merged to a score of 1). The resulting dataset contained levels of class imbalance that varied across the responses, and the ratios of the class imbalances are presented in Section 3.2.

All data processing and analysis was completed using Matlab2019a [28] and all data collection procedures carried out in this study were approved by the University of New England’s Animal Ethics Committee (AEC 16-087).

### 2.2. Principal Component Analysis of the Feature Set

In order to understand the structure and relationship of correlations within the dataset, Principal Component Analysis (PCA) was applied to the feature set outlined in Section 2.1. Z-Score normalization was applied to each of the features to remove the effect of scale from the analysis. Principal Component Analysis is a common dimensionality reduction technique that can be applied to complex, high-dimensional dataset to generate a more easily interpreted low-dimension representation. PCA works by projecting data set onto lower-dimensional hyperplanes so that the largest proportion of the variance is preserved. This process is carried out iteratively to calculate projections for the desired number of principal components, where each subsequent hyperplane projected at right angles to a previous hyperplane [35] (Jolliffe, 2003). The creation of this lower dimensional representation can help uncover complex correlations between features by analyzing how each feature maps to the principal component hyperplanes. The PCA implementation used in this study was provided by the MATLAB Statistics and Machine Learning toolbox [36]. The resulting principal component coefficients and scores were then plotted on a biplot to provide a visual representation of the correlations within the data. The biplot provides a two-dimensional representation of the data with vectors plotted (one for each feature) representing the weighting that each feature has on the first two principal components. The distance from the origin provides a measure of the feature’s contribution to the variance within a principal component analysis (i.e., the larger the vector, the more influence the feature has on the principal component). The angles between the vectors represent the correlations between the corresponding features. Small angles (i.e., θ ≈ 0°) between two vectors indicate that the corresponding features are likely to be positively correlated. If the vectors meet at right angles (i.e., θ ≈ 90°), the features are likely uncorrelated. If the vectors have large angles (i.e., θ ≈ 180°) between them, they are likely to be negatively correlated.

### 2.3. Unbalanced Data and Minority Analysis

One of the key challenges with the dataset is the level of imbalance between the positive and negative examples for the set of response variables. Unbalanced datasets occur when some classes in the distribution significantly dominate others. This makes training difficult because most binary classification algorithms assume an equal cost for misclassification on both classes, leading the learner to correctly classify almost all of the majority class at the expense of misclassifying the most of the minority class [37,38]. As mentioned in Section 2.1, all response variables have been simplified to consist of binary classes, however, it was evident there is a significant imbalance between the minority and majority classes. The approach outlined by [39] was used to analyze the nature of the class imbalance, whereby each data point was classified as safe, borderline, rare, or outlier based on achieving ratios of 5:0/4:1, 3:2/2:3, 1:4, and 0:5, respectively, for the minority-to-majority point ratio across the five nearest neighbors. In this scheme, a safe data point is likely to be learned by the model, whereas an outlier data point is unlikely to be learned (especially if not represented in the training partition). Borderline and rare data points will result in mixed performance depending on the nature of the model. The proportion of each data point type for each response variable was then plotted for comparison across the responses. To visualize the dataset and further demonstrate the nature of the class imbalance, t-Distributed Stochastic Neighbor Embedding (t-SNE) run with default parameters [40] was applied to the data to produce a two-dimensional representation of the 16-feature dataset. The resulting two-dimensional dataset was then plotted with classes denoted using visual markers to visually assess the relationship between the minority and majority points.

### 2.4. Machine Learning Workflow

To investigate the application of machine learning to the RFID data, the random forest machine learning algorithm was applied to the dataset [41]. The random forest is an ensemble machine learning algorithm that aggregates its classifications from the results of multiple diverse decision trees (referred to as weak learners) that are constructed by selecting the features used within each split to form a random subset of all training features. The individual weak learners are not very accurate at correctly classifying outcomes on their own, however when their outcomes (i.e., the probability of belonging to a given class) are averaged, they form a more accurate classifier. The random forest algorithm is well-suited to multi-dimensional classification problems and has been successfully applied across bioinformatics datasets for a range of classification tasks [42,43]. It represents the natural choice for this study as it can handle large and noisy datasets without overfitting, is robust when outliers are present in training data, provides a robust ability to measure feature importance, and is capable of modelling complex non-linear relationships between the input features and the target outputs [44].

The random forest algorithm has several hyper-parameters that were optimized through cross-validation to ensure that the model is fitted to the training data. Within each trial of the machine learning workflow, the number of decision trees (i.e., the size of the ensemble), along with the depth and number of decision splits within the individual trees, was optimized to fit the model to the training set. The number of predictors that were randomly selected for each decision split within each individual decision tree was fixed at 4 (the square-root of the number of features used for training [41]). Feature selection was not performed, and the random forest models were trained using all 16 available features. This allowed for the analysis of feature importance by applying the permutation method, where the values for each feature are permutated and the mean decrease in classification accuracy were used to compare the importance of the features for the classification task [45]. In this approach, a larger mean decrease in accuracy indicates a higher level of importance for the classification task. This process was completed for each feature and provided an indication of the features that had the highest impact on the performance of the classifier.

The workflow used to construct the machine learning models is outlined in Figure 1. It consisted of four stages:

Stage 1: Division of the base dataset into training/validation (75%) and testing (25%) sets using stratified random sampling. The training/validation set was used to train the random forest algorithm and hyper-parameter optimization, while the testing set was used for an out-of-sample test for the predictive power of each model. The datasets were z-score normalized after the separation into these respective sets to present information leaking from the test set into the training set.

Stage 2: Balancing the training set to ensure that the model achieved the most balanced sensitivity to the minority and majority classes. In this investigation, the kernel-smoothed bootstrap sampling approach [46] was adopted for this purpose.

Stage 3: Training and validation. This involved systematically testing parameter configurations to find the configuration with the highest performance when tested against the validation set.

Stage 4: Testing against the out-of-sample testing set put aside during stage 1. This provided a solid indication of the predictive performance that could be achieved on future (novel) data points. In order to assess the performance, the area under the ROC curve, accuracy (Equation (1)), sensitivity (Equation (2)), and precision (Equation (3)) for the classification of the minority class were calculated.
(1)$$\mathrm{Predictive}\;\mathrm{Accuracy}=\frac{TP+TN}{TP+TN+FP+FN}$$
(2)$$\mathrm{Sensitivity}=\frac{TP}{TP+FN}$$
(3)$$\mathrm{Precision}=\frac{TP}{TP+FP}$$
where ***TP*** is the number of true positives (actual positives classified as positives), ***TN*** is the number of true negatives (actual negatives classified negatives), ***FP*** is the number of false positives (actual negatives classified as positives), and ***FN*** false negatives (actual positives that are classified as negatives). Boxplots were generated to present the distributions of each performance variable with the bottom and top box edges indicating the 25th and 75th percentiles, respectively, and the red center lines representing the median. Outliers (denoted by red crosses) are defined as any value that is more than 1.5 times the interquartile range outside the box edges.

This workflow was completed across 1000 trials using different randomized divisions (in stage 1) within each trial to provide a distribution for each of the performance measures for each of the responses tested. Random forest models were also created for random permutations of each response variable within each trial. This provided a randomized control data set that maintained the same ratio of minority-to-majority examples so that the performance of classifiers trained on the real response data could be compared against that of the random response (i.e., the random response should have an expected AUC ≈ 0.5). The distributions of AUC performance metrics for the classifiers trained on the random permuted responses and the real responses were compared using a two-sample Kolmogorov–Smirnov test with rejection of the null hypothesis (that the two distributions are the same) tested at the 5% significance level. This provided a method to assess the performance of the models in marginal cases where the class imbalance made training the models almost impossible and performance was poor.

## 3. Results

### 3.1. PCA Feature Analysis

The initial study of the underlying correlations within the PCA analysis of the feature set is presented in Figure 2. The biplot in Figure 2 (left) shows the contributions of the 16 features to the variance in the first two principal components. This plot shows that the mean durations for the given zones in the system were correlated across the different production periods. For example, the small angles between the vectors representing the mean durations spent at the upper feed (UF) across the PRL, LL, EL, and PL indicate that they are all correlated. This same trend is evident for the nest box (NB), range (RNG), and lower feeder (LF). The UF mean durations were negatively correlated with the mean durations for the RNG and LF (indicated by the large angles between the respective set of vectors). Interestingly, there was a correlation between LF mean durations and the RNG mean durations. These results indicate that there may be a relationship between the times spent within these three zones in the production system. The mean time spent in the NB was not correlated with any other mean durations, indicating that time spent in this location was independent of that spent in other areas. Figure 2 (right) provides a plot of the explained variance across the first ten principal components. This plot shows that the first two components (presented in the biplot in Figure 2 (left)) accounted for 52.0% of the total variance within the dataset (37.1% in the first component alone) and the first 10 principal components accounted for 93.7% of the total variance within the data.

### 3.2. Unbalanced Data and Minority Analysis

The results from the minority analysis are presented in Figure 3, providing insight into the suitability of the dataset to train predictive classifiers for the tested responses. In Figure 3 (left), the ratio of the minority-to-majority class data points is shown for each response. These ratios ranged from a split of 49:51% for the presence of keel bone fractures right down to 0.7:99.3% for the presence of beak damage, where a significant class imbalance is present. Figure 3 (right) plots the proportion of the minority class data points that fit into each of the nearest neighbor classes (outlier, rare, borderline, and safe), presented in the order specified by the minority-to-majority ratio. Only the two least imbalanced responses (presence of keel bone fractures and presence of cestodes) contain any data points that can be classified as safe. These two responses also have relatively small proportions of outlier points (3% and 7%, respectively) in contrast to the most imbalanced data class (beak damage), which consists of 91% outliers.

The effect of the class imbalance and the large proportions of outlier minority data points are demonstrated visually in Figure 4, which plots the two-dimensional t-SNE projection of the dataset with the responses for keel bone fractures (left) and beak damage (right) (i.e., the most imbalanced and least imbalanced responses). In the keel bone fractures response data, Figure 4 (left), it is evident that there are some areas where minority class data points are concentrated, corresponding to the safe data points within the set. There is still significant dispersion of minority class data points through the entire feature space, which is reflective of the large proportion of rare points. Figure 4 (right) demonstrates a largely outlier dataset where the minority points are not only acutely outnumbered but also dispersed through the dataset, with very few that have same-class neighbors.

### 3.3. Classifier Performance for Predicting the Health Status of Hens

The raw classifier performance results are presented in Table 2, where the mean sensitivity, precision, accuracy, and AUC values for each response (and each class where relevant) are reported along with the standard deviation for each respective measure. These results are elaborated on in Figure 5, where the distributions for the AUC, accuracy along with the sensitivity, and precision for the minority class across each respective response are represented using box plots. The results demonstrate that models for predicting the presence of SLD, *A. galli*, and cestodes performed best with mean AUC values of 0.61, 0.60 and 0.58 respectively. There was a relatively large difference to the next-highest performing model, approaching a very marginal performance close to an AUC of 0.5. These models achieved sensitivities of 0.28, 0.33, and 0.33 for the minority classes, respectively, indicating that they were able to detect about 1/3 of the cases (for these responses, the minority class indicates the presence of the disease or infestation). The mean precisions for these responses varied more widely, with mean values of 0.14, 0.28, and 0.49, respectively. This indicates that, of the data points classified as positive for the SLD/parasite infestation, 14%, 28%, and 48% of them correspond to true positives, respectively. Sensitivity and precision scores for the other responses largely reflect the class imbalance, the majority classes achieve relatively high scores for both, and the minority classes have marginal performance. The net results were reflected in the marginal AUC scores that approach those of a chance model. While the accuracy results have been reported, they needed to be interpreted with care as the 25% testing partitions one which performance testing is based has the same minority:majority class ratio as the original dataset (see Figure 1), introducing a significant imbalance into the calculation. This issue can be clearly seen in performance results for beak damage where a mean accuracy of 0.98 is achieved, however only mean sensitivity of 0.06 with a precision of 0.02 was seen for this minority class. The AUC value for beak damage, which is a much more robust measure in the presence of class imbalance, was only 0.48, and provides a much better indication of this response/model’s overall performance. All AUC distributions presented in Figure 5A were significantly different from the AUC distributions of models trained on random permutations of the responses (*p* ≈ 0), except for the egg follicle response (test-statistic = 0.06, *p* = 0.06). This indicates that its performance is equivalent to a chance classifier.

Figure 6 presents the distributions of the feature importance (mean decrease in accuracy) for 16 features used to train the top 3 performing responses (SLD, *A. galli*, and cestode infestation) by AUC. The distributions were ranked according to their mean and showed that the pre-laying (PRL) mean durations for the upper feeder (UF) and range (RNG) rank first or second across all three responses, with MDAs ≈ 6–7%. The NB (PRL) feature ranked 3rd for SLD and *A. galli* responses and dropped to 9th for the cestodes response. The UF (EL) and UF (LL) features rank in the top 6 of sets for all three responses. Based on these observations, new sets of models were generated using only four PRL features (UF, LF, NB, RNG) as predictors to assess the ability of the pre-laying mean durations alone to predict the responses. The resulting performance distributions for these models are presented in Figure 7. The results presented for the reduced feature-set models reflect those presented in Figure 5, with the SLD, *A. galli*, and cestodes responses achieving the highest performance (by mean AUC), ranging down to marginal results for the remaining responses with mean AUC values below 0.5. The distributions for mean sensitivity (minority class) achieved of three top performing models were slightly higher than those trained on the full feature set, while their mean precision distributions were slightly lower, contributing to a slightly lower overall performance. Figure 8 plots the distributions of the feature importance for the three top-performing responses trained on the PRL features. These results reflect those in Figure 6, with the UF and RNG features ranking first and second for the responses shown. The MDA distributions for the UF and RNG features are shifted up slightly (compared to the distributions in Figure 6), with median values above 7% for the SLD and *A. galli* responses.

## 4. Discussion

The results demonstrate mixed performance across the responses tested, with the highest performing models only achieving a relatively modest overall predictive ability. The three best performing responses in this study (SLD, *A. galli*, cestodes) have all been shown in previous studies to be strongly linked to access to soil and litter [10,11,47]. From this study, it is evident that there is a modest degree of predictive ability within the range/aviary usage patterns that are described by the mean durations. The correlations presented through the PCA (Figure 2) and evidencing through the feature importance results are consistent with the findings in previous work [30,33]. This may indicate that sub-populations within the flocks, consistent of hens that frequently use the range and hens that rarely access the range, are also using different spaces within the hen house, e.g., preferring to use the upper feeder versus the lower feeder.

The mean durations spent on the range and at the upper feeder were the top two ranked features used for prediction in all three top performing responses and the PCA analysis demonstrated that these variables are negatively correlated across the dataset (and range access is correlated with usage of the lower feeder). This indicates that hens that spent a higher mean duration on the range (i.e., with access to soil and litter) spend less time at the upper feeder (away from soil and litter) and vice-versa, resulting in a usage profile across the production lifecycle with a modest level of predictive power. The results in Figure 7A indicate that most of this predictive capability is present within the mean durations from monitoring within the pre-laying period. This is unsurprising, given the observation that the mean durations for specific locations (RNG, UF, LF, and NB) within the system are highly correlated across the monitoring time periods (PRL, PL, LL, and EL). The results demonstrate that monitoring in the early production period provides sufficient sensitivity to identify 1/3 of the cases of hens affected by SLD, *A. galli* infection, or cestode infestations that were ultimately present at the end of the flock (with varying levels of precision). While it is disappointing to observe that monitoring in subsequent production periods does not appear to greatly improve model performance, this introduces the possibility of early and targeted intervention, where preventative treatments applied to individuals or groups with high proportions of birds predicted to be at risk can be adopted to boost the productivity of the flock based upon monitoring in the pre-laying period. While in this study the precision for these three responses was low (14%, 28% and 49% respectively), the cost of applying preventative treatments to the hens detected as false positives may be outweighed by the ultimate production increase achieved over the life of the flock.

The remaining responses (FLHS, keel bone fractures, egg follicle production, comb or wattle damage, and beak damage) all demonstrated marginal performance over a chance classifier. The comb or wattle damage and beak damage responses were highly imbalanced, with their minority classes comprising just 2.5% and 0.7%, respectively. This level of class imbalance coupled with large proportion of outlier minority data point is difficult to overcome to effectively train a machine learning classifier, resulting in poor performance. The effect of the outlier minority data points (a demonstration of the small disjuncts described by [39] is illustrated in Figure 4 (right), where minority data points for the beak damage response are sparse across the data space and removal of any individual point (i.e., for the out-of-sample test) leaves only majority points behind the region. This eliminates any possibility of training a classifier to recognize the presence of minority data within this region. The responses for FLS and egg follicle production had similar ratios of imbalance to *A. galli* and SLD, respectively. Their predictive performance was lower (e.g., AUC and sensitivity to the minority classes), indicating a lack of predictive ability within the mean duration features in the dataset, rather than an effect of the class imbalance. This effect is further demonstrated by the egg follicle production response higher proportion of outlier minority points compared to the better performing responses.

The minority analysis presented within this study provides an elegant demonstration of the role of both minority/majority class ratios and the nature of data within the classes. As the ratio of minority-to-majority class points grows, the general trend shows that the model performance increases, however, this trend is strongly influenced by the proportions of the safe, borderline, rare and outlier points that make up the minority class. Recall that the borderline and rare data points have mixed performance, as they indicate areas within the dataspace where data points of both classes are mixed. Therefore, in order for model performance to be high, more balanced datasets need to have higher proportions of safe points. This is because the absolute number of data points exposed to the lower performing classes of points will be higher (e.g., a more balanced dataset has more points in the minority class). This principle is demonstrated in the difference between performance between the keel bone fracture response (which is almost balanced 49:51) and cestode infestation response (which has a ratio of 39:61). Inspection of the minority class proportions in Figure 3 (right) reveals that although the keel bone fracture response is 10% more balanced overall, it has only 3% more safe data points (7% vs. 4% for the keel bone fractures and cestodes, respectively). This results in a larger absolute number of minority data points falling into the borderline, rare and outlier classes, where classification for performance is lower. This is subsequently reflected in the overall performance, where the cestode infestation response achieves a higher mean AUC and mean sensitivity while maintaining a similar mean precision to the keel bone fracture response.

One of the key shortcomings present within the structure of the data is the measurement of the response variables at the end of production (i.e., through necropsy). This means that there is no indication within the data, at what time point during the 74-week production period a particular hen became affected by any of the health conditions described through the response variables. This is a likely source of noise within the dataset, introducing a number of different feeder/range/nest box usage patterns across the measurement periods for a given response based on when a specific bird becomes affected by a condition. Further investigation is needed to produce a more refined training set that captures the usage patterns immediately prior to the diagnosis of a response. Predictive performance may also be further improved by the incorporation of additional data (e.g., weather climatic conditions, aggregate feed intake, and egg production measures from the production system). These measures have the potential to provide a more complete view of flock health in combination with the individual-level system usage pattern collected through RFID technology. This study is based upon the random forest ensemble algorithm with a robust classifier that performs well on a wide variety of datasets and is resistant to overfitting data. The trained models are complex (e.g., ensembles are constructed within multiple diverse weak learner models), especially when trained on the full feature set. Future studies should investigate the use of less complex algorithms and the implementation of a complete feature selection component within the workflow. Approaches such as Recursive Feature Elimination [48] and Infinite Feature Selection [49] provide examples of supervised and unsupervised approaches that could be applied to simplify the feature set and train robust, simplified models using other machine learning approaches.

## 5. Conclusions

The study presents a novel analysis using machine learning to predict the presence of a set of health parameters from RFID data measuring the average durations spent within key locations in the production system. The key findings indicate that the presence of SLD, *A. galli*, and cestodes may be predicted based upon flock monitoring in the early (pre-laying) period of the production cycle. However, the predictive power is modest, and additional analysis (such as the incorporation of additional data streams) is needed to improve the approach for a practical system. Range access and upper-feeder was shown to be key predictors for SLD, *A. galli*, and cestodes infections. The minority analysis demonstrated a significant imbalance with many of the minority points isolated within the feature space, making it difficult to fit an effective model for these responses. Additional minority class data points would help solve this problem by increasing the density and number of safe minority points. The ability to predict the presence of other health conditions was limited and no practical outcomes were evident. The machine learning workflow, incorporating analysis of the class imbalance, the application of a sampling algorithm, and the refinement of predictive data features provides a sound model for understating the data and the performance outcomes achieved. This approach can be applied in future studies to understand the nature and effects of class imbalance on machine learning performance. Overall, the study provides a demonstration of the potential for RFID movement data based on individual animals and machine learning approaches to predict health outcomes. We believe that our study lays a foundation for future research in this area and that our findings can inform the development of more accurate and effective health monitoring methods for laying hens.

## Figures and Tables

**Figure 1 animals-13-01202-f001:**
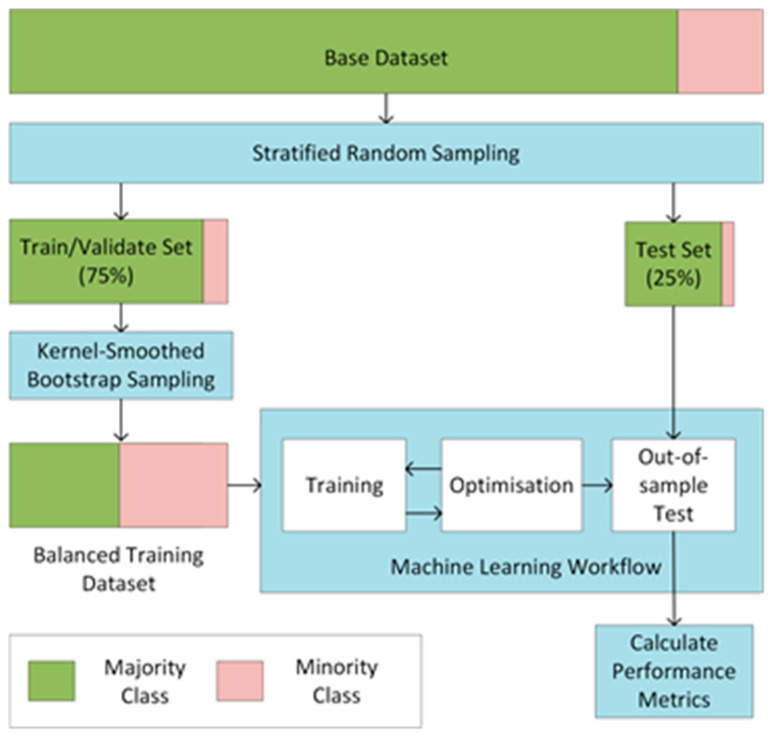
The outline for the machine learning workflow adopted demonstrating the data partitioning and minority class re-sampling process. The blue areas denote processes in the workflow. The Kernel-Smoothed bootstrap sampling was performed after the testing set was partitioned from the train/validate set to ensure a complete out-of-sample test.

**Figure 2 animals-13-01202-f002:**
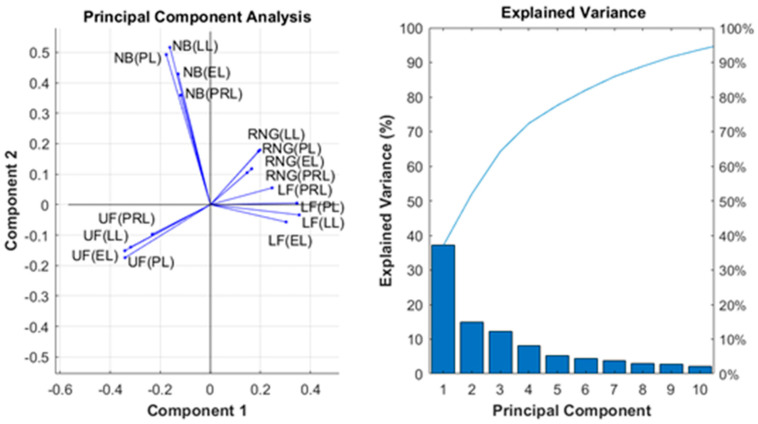
(**Left**) Principal Component Analysis (PCA) across all 16 features visualized using a biplot normalized to arbitrary units. UF, LF, RNG, and NB notation refers to the mean durations spent at the upper feeder, lower feeder, range, and nest box, respectively, for each of the denoted productions periods (pre-lay = PRL, peak lay = PL, late lay = LL, and end of lay = EL). (**Right**) A Pareto chart plotting the proportion of explained variance within each component. The blue line plots the cumulative variance across the components from left to right.

**Figure 3 animals-13-01202-f003:**
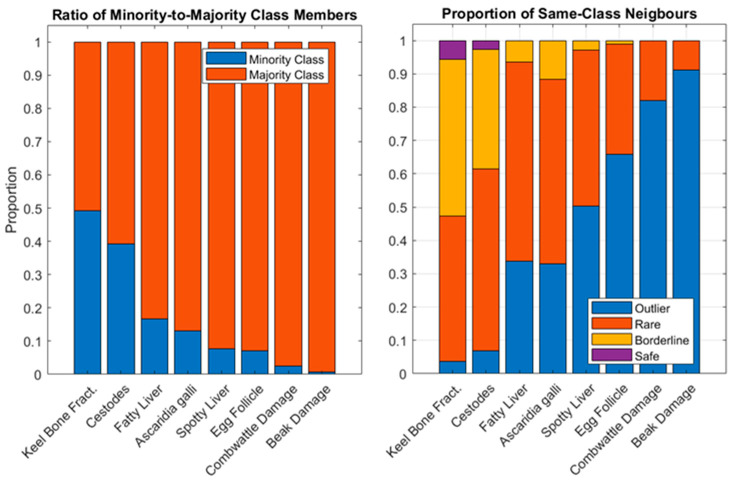
(**Left**) the ratio of minority-to-majority class samples for each target. (**Right**) the proportion of data points in the minority class that fall into the outlier, rare, borderline, and safe neighbor categories.

**Figure 4 animals-13-01202-f004:**
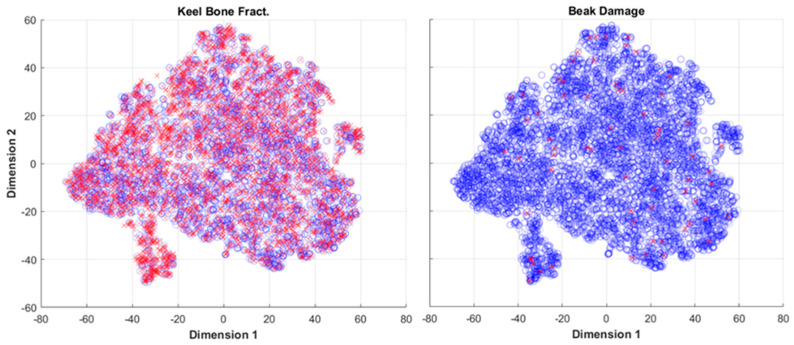
Two-dimensional representation of the dataset constructed using t-SNE for (**left**) the most balanced data set, keel bone damage, and (**right**) least balanced dataset, the beak damage, response with the minority class point denoted with red crosses and majority class points designated with blue circles.

**Figure 5 animals-13-01202-f005:**
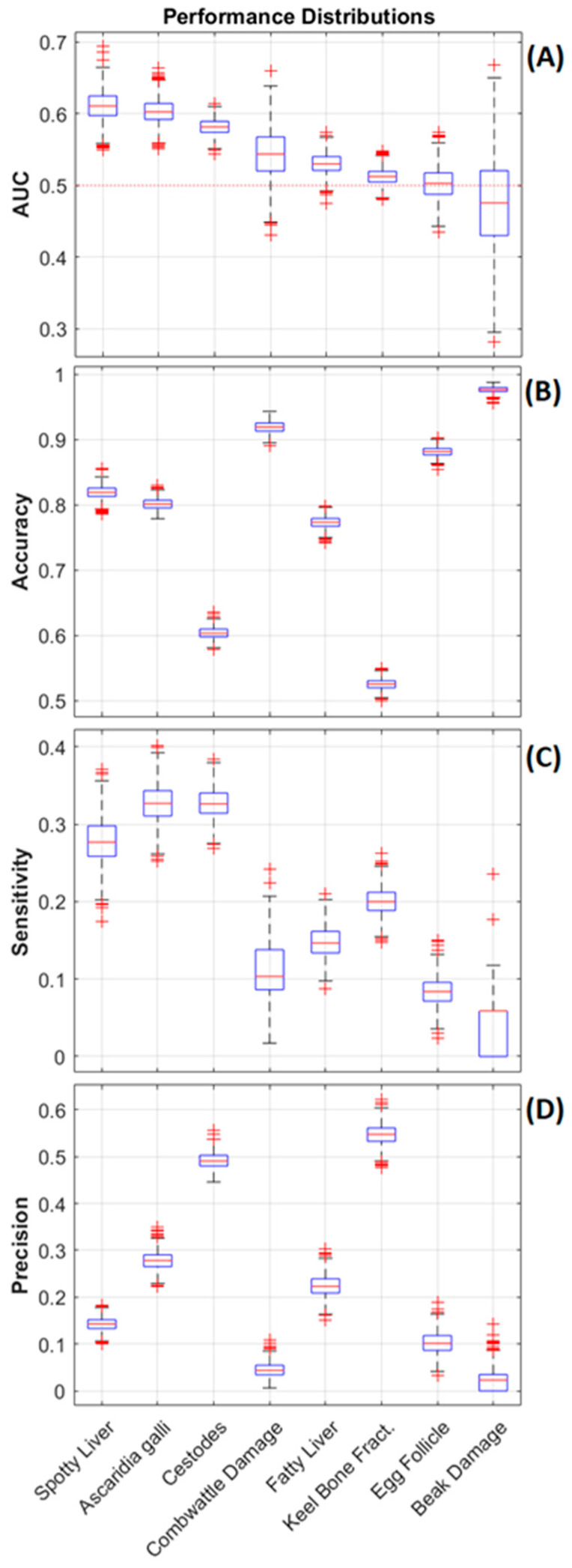
Performance distributions plotted as boxplots for each response variable. Part (**A**) plots the area under the ROC curve (AUC), with the red dotted line marking the performance of a random-chance classifier. Part (**B**) plots the accuracy for each model. Part (**C**) plots the sensitivity, and part (**D**) plots the precision for the minority class within each response. The responses are ordered from left to right by descending mean AUC.

**Figure 6 animals-13-01202-f006:**
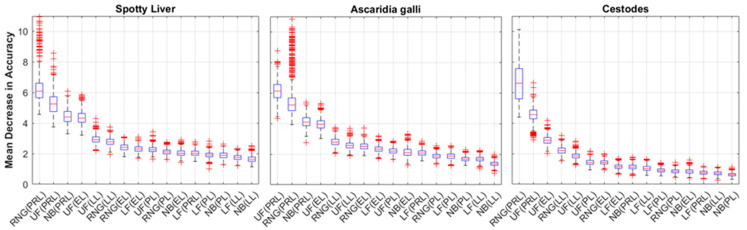
Feature importance measured through the mean decrease in accuracy for the three responses with the highest AUC values—(**left**) Spotty Liver Disease, (**middle**) *Ascaridia galli*, and (**right**) Cestode infections. The responses have been classified according to the location of the hen on either the range (RNG), upper feeder (UF), lower feeder (LF), or nest box (NB) during pre-lay (PRL), peak lay (PL), late lay (LL), or end of lay (EL).

**Figure 7 animals-13-01202-f007:**
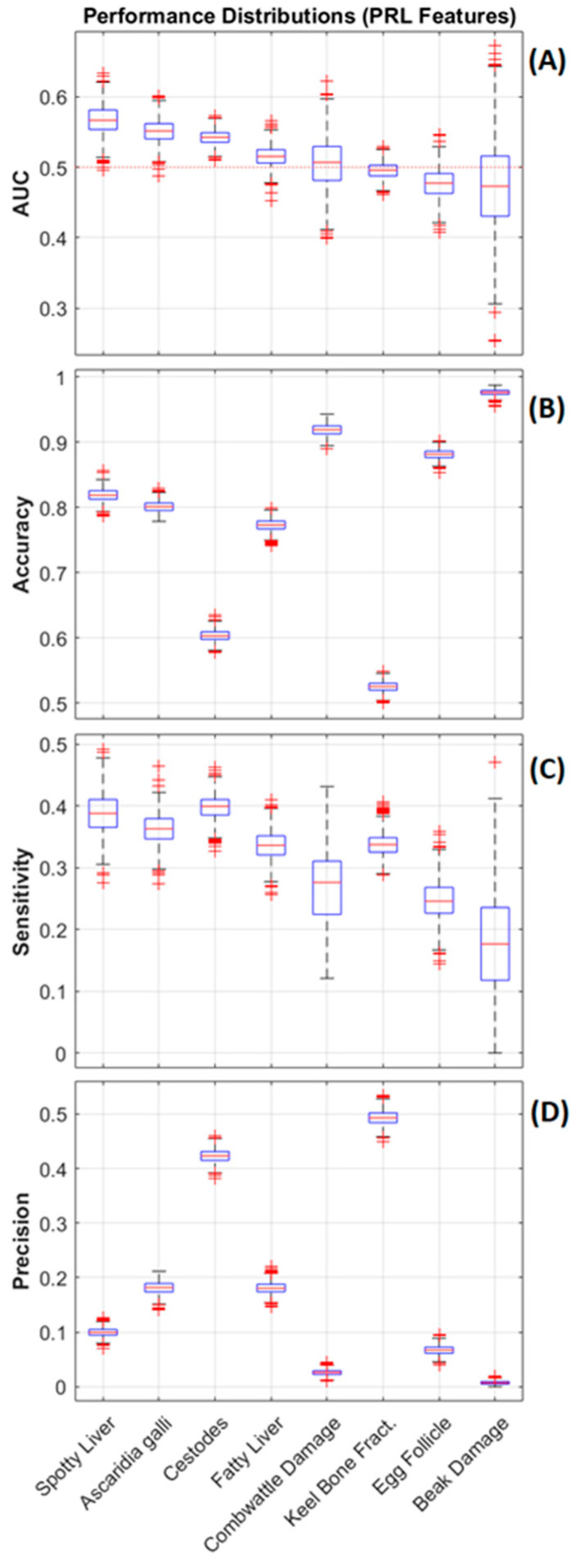
Performance distributions plotted as boxplots for each model using only the PRL features. Part (**A**) plots the area under the ROC curve (AUC), with the red dotted line marking the performance of a random-chance classifier. Part (**B**) plots the accuracy for each model. Part (**C**) plots the sensitivity, and part (**D**) plots the precision for the minority class within each response. The responses are ordered from left to right by descending mean AUC.

**Figure 8 animals-13-01202-f008:**
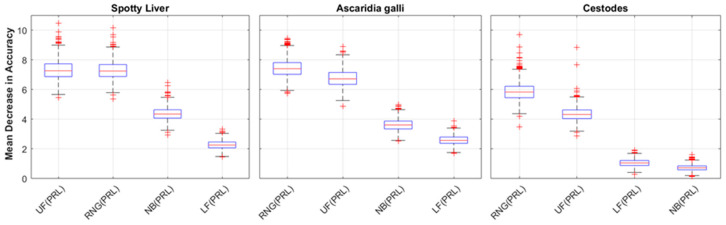
Feature importance measure through the mean decrease in accuracy for the three responses with the highest AUC values—(**left**) Spotty Liver, (**middle**) *Ascaridia galli*, and (**right**) Cestode infestation. The responses have been classified according to the location of the hen on either the range (RNG), upper feeder (UF), lower feeder (LF), or nest box (NB) during pre-lay (PRL), peak lay (PL), late lay (LL), or end of lay (EL).

**Table 1 animals-13-01202-t001:** The definition of the response/target variables and classes used for the machine learning models.

Target	Response Value Definition
Keel bone damage	0 = No damage, 1 = signs of damage ranging from minor to severe fractures
Fatty Liver Hemorrhagic Syndrome	0 = Normal physiological liver, 1 = evidence of Fatty Liver Hemorrhagic Syndrome
Spotty Liver Disease	0 = Normal physiological liver, 1 = evidence of Spotty Liver Disease
Presence of *Ascaridia galli*	0 = Not present, 1 = Present
Presence of Cestodes	0 = Not present, 1 = Present
Egg follicle production	0 = No active follicles or follicles in regression, 1 = Full follicle production
Beak damage	0 = Damaged beak, 1 = No damage
Comb or wattle damage	0 = Damaged comb or wattle, 1 = No damage

**Table 2 animals-13-01202-t002:** Performance results for the random forest classifiers trained on each response. The mean sensitivity, precision, accuracy, and Area Under the Curve (AUC) (± standard deviation) are presented along with the number (N) of data points in each class.

Response	Class	N	Sensitivity	Precision	Accuracy	AUC
Keel bone fractures	0	4605	0.20 ± 0.02	0.55 ± 0.02	0.52 ± 0.01	0.51 ± 0.01
	1	4757	0.84 ± 0.02	0.52 ± 0.00		
Egg follicles	0	670	0.08 ± 0.02	0.10 ± 0.02	0.88 ± 0.01	0.50 ± 0.02
	1	8686	0.94 ± 0.01	0.93 ± 0.00		
Cestodes	0	5693	0.78 ± 0.02	0.64 ± 0.01	0.60 ± 0.01	0.58 ± 0.01
	1	3669	0.33 ± 0.02	0.49 ± 0.02		
*Ascaridia galli*	0	8136	0.87 ± 0.01	0.90 ± 0.02	0.80 ± 0.01	0.60 ± 0.02
	1	1226	0.33 ± 0.02	0.28 ± 0.00		
Spotty Liver Disease	0	8651	0.86 ± 0.01	0.94 ± 0.00	0.82 ± 0.01	0.61 ± 0.02
	1	711	0.28 ± 0.03	0.14 ± 0.01		
Fatty Liver Hemorrhagic Syndrome	0	7803	0.90 ± 0.01	0.84 ± 0.00	0.77 ± 0.01	0.53 ± 0.01
	1	1559	0.15 ± 0.02	0.22 ± 0.02		
Beak damage	0	68	0.06 ± 0.05	0.02 ± 0.02	0.98 ± 0.00	0.48 ± 0.07
	1	9294	0.98 ± 0.00	0.99 ± 0.00		
Comb or wattle damage	0	9129	0.94 ± 0.01	0.98 ± 0.00	0.92 ± 0.01	0.54 ± 0.03
	1	233	0.11 ± 0.04	0.05 ± 0.02		

## Data Availability

The data presented in this study are available on request from the corresponding author.

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
