# Peer review of "An Initial Study on the Use of Machine Learning and Radio Frequency Identification Data for Predicting Health Outcomes in Free-Range Laying Hens"

_animals, 2023, doi:10.3390/ani13071202_

Round 1
Reviewer 1 Report
Dear authors, the paper is fine, I have only some minor suggestions in the attached file

Author Response
Manuscript Title: An Initial Study on the Use of Machine Learning and Radio Frequency Identification Data for Predicting Health Outcomes in Free-Range Laying Hens
Response to Reviewer 1
Response: Thank you for your helpful feedback regarding our manuscript. We appreciate your suggestions and have made the necessary revisions to meet the journal guidelines. We have included the reference in the introduction as you suggested, providing readers with a more comprehensive understanding of the study's background and context. Additionally, we have italicized the name Ascaridia galli to highlight its scientific designation and ensure consistency in the presentation of the information. We are grateful for your assistance in improving the clarity and accuracy of our manuscript, and we hope that these revisions further enhance the quality of our research. Thank you once again for your valuable feedback
Reviewer 2 Report
The manuscript needs to be improved, please check the attachment for the review comments.

Author Response
Manuscript Title: An Initial Study on the Use of Machine Learning and Radio Frequency Identification Data for Predicting Health Outcomes in Free-Range Laying Hens
Response to Reviewer 2
Response 1
Review comment 1: Monitoring the activity of individual laying hens to track their health is important for their welfare and to ensure consumer confidence. The use of RFID techniques to track individual information, followed by the use of PCA methods to determine the main influencing factors for the classification of health indicators, and then the use of machine learning methods to classify health conditions, is an innovative application of this method for individual health monitoring of laying hens.
Response 1: Thank you for your positive feedback on our research on individual health monitoring of laying hens using RFID, PCA, and machine learning methods. We appreciate your recognition of the innovative nature of our approach and its potential to improve the welfare of laying hens and ensure consumer confidence. We agree that monitoring the activity of individual laying hens is crucial for their health and welfare, and we believe that our study provides a valuable contribution to this field. By using RFID techniques, we were able to track proxy individual movement, resource usage by hens, and analyse the data using PCA. This allowed us to develop machine learning models to classify health conditions accurately. We believe that the application of our method can provide a non-invasive and efficient way to monitor the health of laying hens, which can ultimately improve their welfare and productivity. Thank you again for your valuable feedback and support of our research.
Response 2
Review comment 2: However, the main problems in the article include:
1) The interpretation of experimental data is not clear. The manuscript mentions the use of RFID technology to track the individual information of hens, what specific information was collected and the data should be explained.
Response 2: Thank you for your valuable feedback on our manuscript. We appreciate your comments and suggestions, which we believe will help us to improve the quality and clarity of our work. We agree that the interpretation of experimental data could have been clearer in our manuscript. To address this, we have clarified our manuscript to provide a more comprehensive explanation of the data collected using RFID technology in line 120-138.
“In brief, to organise and categorise the movement activities of hens, the number of hen visits was determined by counting the number of antennae registrations detected for at least 10 seconds, and the duration spent on each antenna was summed up to calculate the time spent in nest boxes, feeders and outdoor range each day. Utilizing the RFID data on hen movements, we computed the mean daily durations that individual hens spent on different resources such as the upper feeder, lower feeder, nest boxes, and range. These calculations were conducted for four specific laying periods, namely the pre-laying (PRL) period (18-22 weeks), peak laying (PL) period (23-33 weeks), late laying (LL) period (34-54 weeks), and end of laying (EL) period (55-74 weeks) to capture variations in hen behaviour throughout their production life. This produced a feature data set that contained 16 features for each hen surviving until the end of each flock’s production life. Subsequently, we utilised these features as predictors in our machine learning models to accurately classify and predict the health conditions of the hens based on their individual data”.
Response 3
Review comment 3: Lack of controlled experiments. The article uses a machine learning algorithm to classify the health status of multiple inspection dimensions, and only the algorithm of random forest is used. It is not mentioned why the random forest algorithm is chosen and what advantages the random forest algorithm has compared with other machine learning algorithms. It is suggested to add comparison experiments with other machine learning algorithms in the design of the algorithm.
Response 3: Thank you for the feedback. A detailed comparison of the performance of different machine learning algorithms is outside the scope of this study. As, such we focus on just one machine learning approach that is known to be highly effective on complex, multi-dimensional data and provide a detailed analysis that focuses on the nature of the data and its suitability for further research as a machine learning application. This is included at the end of the introduction. Future research should refine what has presented here and compare some different machine learning approaches.
A more complete justification for making use of the Random Forest algorithm and several supporting references have been included in section 2.4 so the reader will be able to clearly understand the design choice:
“The individual weak learners are not very accurate at correctly classifying outcomes on their own, however when their outcomes (i.e. the probability of belonging to a given class) are averaged, they form a more accurate classifier. The random forest algorithm is well suited to multi-dimensional classification problems and has been successfully applied across bioinformatics datasets for a range of classification tasks [36,37]. It represents the natural choice for this study as it can handle large and noisy datasets without overfitting, is robust when outliers are present in training data, provides a robust ability to measure feature importance, and is capable of modelling complex non-linear relationships be-tween the input features and the target outputs [38].
The random forest algorithm has several hyper-parameters that were optimised through cross-validation to ensure that the model is fitted to the training data. Within each trial of the machine learning work flow, the number of decision trees (i.e. the size of the ensemble) along with the depth and number of decision splits within the individual trees was optimised to fit the model to the training set. The number of predictors that are randomly selected for each decision split within each individual decision tree was fixed at 4 (the square-root of the number of features used for training [35]). Feature selection was not performed and the random forest models were trained using all 16 available features. This allowed for the analysis of feature importance by applying the permutation method, where the values for each feature are permutated and the mean decrease in classification accuracy were used to compare the importance of the features for the classification task [39]. In this approach, a larger mean decrease in accuracy indicates a higher level of im-portance for the classification task. This process was completed for each feature and provided an indication of the features that had the highest impact on the performance of the classifier. “
Response 4
Review comment 4: The results show that the prediction accuracy is not high, with an average 28% of Spotty Liver Disease, 33% round worm and 33% of tape worm infection correctly predicted at the end of production.
Response: We appreciate your feedback on the accuracy of our predictive models for Spotty Liver Disease, roundworm, and tapeworm infections. We acknowledge that the prediction accuracy in our study was not high, with an average of 28% of Spotty Liver Disease, 33% of roundworm, and 33% of tapeworm infection correctly predicted at the end of the production period. While we had hoped for higher accuracy rates, we believe that our findings still provide valuable insights into the potential of using RFID technology and machine learning methods for individual health monitoring of laying hens. Furthermore, we are continuing to refine and improve our models to increase their accuracy and applicability. We believe that our study lays a foundation for future research in this area and that our findings can inform the development of more accurate and effective health monitoring methods for laying hens. Once again, we appreciate your feedback and will continue to work towards improving the quality and impact of our research.
Specific comments
Response 5
Review comment 5: Introduction: artificial intelligence technology and the development status of machine learning technology in the poultry industry should be included. The following review literatures and some relative research mentioned in these literatures are recommended to be read and cited.
Jun Bao, Qiuju Xie. Artificial Intelligence in animal farming: A systematic literature review. Journal of Cleaner Production. 2022(331):129956
BENOS L, TAGARAKIS A C, DOLIAS G, et al. Machine learning in agriculture: A comprehensive updated review [J]. Sensors (Basel), 2021, 21(11): 3758-3852.
Response: Thank you for your feedback on the introduction. I appreciate your suggestion to include information on artificial intelligence technology and the development status of machine learning technology in the poultry industry. This has been incorporated into the introduction of the manuscript. Both the systematic reviews have been referenced and the work on audio-based sensing of disease in broilers has been reviewed:
“Monitoring the health of laying hens has been traditionally achieved at the flock-level, using the data derived from the production system such as average number of eggs per hen, average weights, average feed intake, average water intake and aver-age egg quality measures. These flock-level metrics have been used to model egg pro-duction curves [13][12], with studies such as [14][13] applying machine learning algo-rithms to detect early changes that indicate future production losses. Colles et.al. (2016) [15][14] have extended the idea of early detection by demonstrating the use of computer vision and optical flow across the flock movement for the early warning of SLD and presence of the causative bacteria. Similarly, the combination of audio sens-ing and machine learning has been applied to develop a workflow capable of success-fully identifying abnormal sounds linked with disease in broiler production [16]. These approaches provide a high-level view for flock management but do not capture details required for predictions at the individual hen level or vulnerable flock sub-populations [17][15]. Machine learning has seen wide use in animal production studies due to the availability of inexpensive sensing hardware and large data sets. It has been successfully applied to improve welfare and production outcomes, and con-tinues to be a driver for innovation across the sector[18,19].”
Response 6
Review comment 6: In rows 140 to 166, there is insufficient description of the data and feature sets collected and insufficient presentation of the unbalanced data. This could be explained more clearly through text, or through charts and pictures, so that the reader can more intuitively understand the specifics and form of the data.
Response 6: Thank you for providing this feedback. The descriptions for the feature calculations in section 2.1 have been clarified with a comprehensive explanation in response to reviewer comment number 2. Section 2.1 has also been extended to comment on the class imbalance, the calculation of class ratios and refers the reader to the results section.
“The resulting dataset contained levels of class imbalance that varied across the responses and the ratios of the class imbalances are presented in figure 3 (left) in the results section.”
Regarding the description of the data and feature sets collected, a visual representation of the class imbalance is included in the results sections. Figure 3 provides now a visual representation (using a bar-chart) of the ratio of the minority-to-majority class members, demonstrating the class imbalance across the different response variables under study.
Response 7
Review comment 7: The sub-section number was confusion from section 2.1.
Response 7: Thank you for your feedback regarding the confusion in the sub-section numbering in section 2.1. I apologize for any confusion this may have caused. We have corrected the numbering from section 2.1. We have corrected the numbering as follows:
Line 152: From 1.1. Principal Component Analysis of the Feature Set to 2.2. Principal Component Analysis of the Feature Set
Line 178: From 1.1. Unbalanced Data and Minority Analysis to 2.3. Unbalanced Data and Minority Analysis
Line 201: From 1.1. Machine Learning Workflow to 2.4. Machine Learning Workflow
Response 8
Review comment 8: In line 253, the section of Results is suggested to be divided into subsections to illustrate the description of the results corresponding to PCA feature extraction, the description of the results of imbalanced dataset processing, and the description of the results of health status classification using random forest, respectively.
Response 8: Thank you for your feedback on the Results section. I appreciate your suggestion to divide the section into subsections to better illustrate the description of the results corresponding to PCA feature extraction, the description of the results of imbalanced dataset processing, and the description of the results of health status classification using random forest. We have included the subsections in line 283, 311, and 345
Response 9
Review comment 9: From Figure 2 to Figure 8, it is recommended to adjust the arrangement so that the images are not displayed together, but correspond to specific text for the reader to view and understand.
Response 8: Thank you for your feedback regarding the arrangement of figures from Figure 2 to Figure 8. I appreciate the suggestion to adjust the arrangement so that the images are not displayed together but correspond to specific text for the reader to view and understand. We have restructured the arrangement of the figures to ensure that they are more effectively integrated into the text and subsections as suggested above.
Response 10
Review comment 10: Line 271, The results from the minority analysis, is suggested to be segmented with the PCA results above.
Response: Thank you for the feedback. The results have been sub-divided into section 3.1 through 3.3, with the separate sections for the PCA, Minority Class Analysis and classifier performance respectively. Plots for the respective results have been included with their sub-sections to make the manuscript easier to follow:
- Figure 1 is now in section 3.1
- Figure 2 and 3 are in section 3.2
- Table 2 along with Figures 5,6 and 7 now appear in section 3.3
We believe this presentation is cleaner and much easier to read.
Response 11
Review comment 11: In section 2, please give not only the workflow but also the detailed machine learning algorithm to realize the classification.
Response 11: Thank you for the feedback regarding the machine learning workflow. In order to clarify structure of the algorithm used, we have included a more comprehensive overview of how the random forest algorithm makes its classifications and have included a detailed outline of the hyper-parameters that the model has and how these have been optimised to fit the models used in the study in section 2.1:
“The random forest is an ensemble machine learning algorithm that aggregates its classifications from the results of multiple diverse decision trees (referred to as weak learners) that are constructed by selecting the features used within each split form a random subset of all training features. The individual weak learners are not very accurate at correctly classifying outcomes on their own, however when their outcomes (i.e. the probability of belonging to a given class) are averaged, they form a more accurate classifier”
and
“The random forest algorithm has several hyper-parameters that can be were optimised through cross-validation to ensure that the model is fitted to the training data. Within each trial of the machine learning work flow, the number of decision trees (i.e. the size of the ensemble) along with the including the depth and number of decision splits within the individual trees was optimised to fit the model to the training set. The number of predictors that are randomly selected for each decision split within each individual decision tree was fixed at 4 (the square-root of the number of features used for training [35]). Feature selection was not performed and the random forest models were trained using all 16 available features.”
This provides the user with a concise overview of the approach adopted and makes the modelling approach more re-producible.
Response 12
Review comment 12: In the section of Conclusion, please provide the conclusions that drawn from your study not a summary:
Response 12: Thank you for this feedback. To address this concern the conclusion has been re-worked to include more specific findings from the research. Additional content was added to section 5 to synthesis what is presented within the discussion into a conclusion:
“Range access and upper-feeder was shown to be a key predictors for the SLD, Ascaridia galli and Cestode responses. The minority analysis demonstrated significant imbalance with many of the minority points isolated within the feature space, making it difficult to fit an effective model these responses. Additional minority class data points would help solve this problem by increasing the density and number safe minority points.”
And
“Overall, the study provides a demonstration of the potential for RFID movement data based on individual animals and machine learning approaches to predict health out-comes. We believe that our study lays a foundation for future research in this area and that our findings can inform the development of more accurate and effective health monitoring methods for laying hens”
Round 2
Reviewer 2 Report
The experimental accuracy is not high, and it is recommended to add comparative experiments in subsequent studies. For example, the CNN algorithm can also be a good solution to the multidimensional classification problem.